# Biochemical Characterization and Functional Analysis of Glucose Regulated Protein 78 from the Silkworm *Bombyx mori*

**DOI:** 10.3390/ijms24043964

**Published:** 2023-02-16

**Authors:** Yao Xiao, Lujie Ren, Yanan Wang, Huanhuan Wen, Yongqiang Ji, Chenshou Li, Yangqing Yi, Caiying Jiang, Qing Sheng, Zuoming Nie, Qixiang Lu, Zhengying You

**Affiliations:** 1College of Life Sciences and Medicine, Zhejiang Sci-Tech University, Hangzhou 310018, China; 2Zhejiang Provincial Key Laboratory of Silkworm Bioreactor and Biomedicine, College of Life Sciences and Medicine, Zhejiang Sci-Tech University, Hangzhou 310018, China; 3Engineering Research Center for Eco-Dyeing and Finishing of Textiles, Ministry of Education, Zhejiang Sci-Tech University, Hangzhou 310018, China

**Keywords:** silkworm, *BmGRP78*, Molecular chaperones, ER-stress

## Abstract

The glucose regulated protein (GRP78) is an important chaperone for various environmental and physiological stimulations. Despite the importance of GRP78 in cell survival and tumor progression, the information regarding GRP78 in silkworm *Bombyx mori* L. is poorly explored. We previously identified that GRP78 expression was significantly upregulated in the silkworm *Nd* mutation proteome database. Herein, we characterized the GRP78 protein from silkworm *B. mori* (hereafter, BmGRP78). The identified BmGRP78 protein encoded a 658 amino acid residues protein with a predicted molecular weight of approximately 73 kDa and comprised of two structural domains, a nucleotide-binding domain (NBD) and a substrate-binding domain (SBD). *BmGRP78* was ubiquitously expressed in all examined tissues and developmental stages by quantitative RT-PCR and Western blotting analysis. The purified recombinant BmGRP78 (rBmGRP78) exhibited ATPase activity and could inhibit the aggregating thermolabile model substrates. Heat-induction or Pb/Hg-exposure strongly stimulated the upregulation expression at the translation levels of BmGRP78 in BmN cells, whereas no significant change resulting from BmNPV infection was found. Additionally, heat, Pb, Hg, and BmNPV exposure resulted in the translocation of BmGRP78 into the nucleus. These results lay a foundation for the future identification of the molecular mechanisms related to GRP78 in silkworms.

## 1. Introduction

Cells experience endoplasmic reticulum (ER) stress when the efficiency of secretory protein folding is threatened, which elicits the homeostatic unfolded protein response (UPR) [1]. Cellular stressors, including toxic chemicals, oxidative stress, Ca^2+^ depletion, and inflammation, can disrupt protein homeostasis and result in the formation of intermediate and misfolded species [2]. High concentrations of mis-folding or aggregating species can lead to cytotoxic complexes in the native cellular environment. Consequently, cells have evolved a complex network of mechanisms to prevent protein aggregation and maintain protein homeostasis (proteostasis) [3,4]. 

Glucose-regulated protein 78 (GRP78), also termed as BiP or HSPA5, is an endoplasmic reticulum (ER) chaperone belonging to the heat shock protein (HSP) 70 family [5]. In contrast to the cytosolic counterparts of the HSP70 protein family, the N-terminal of GRP78 contains a signal peptide targeting it to the ER, and its C-terminal contains the ER retention signal motif KDEL. As an ER resident protein, GRP78 is well known for facilitating UPR under conditions of ER stress, preventing intermediates from aggregation and targeting misfolded proteins for degradation, allowing cells to adapt to adverse stress conditions targeting the ER [6]. The three major ER-transmembrane sensor proteins of the ERS induced UPR pathway: PRKR-like ER kinase (PERK), inositol-requiring enzyme 1 (IRE1), and activating transcription factor 6 (ATF6) are involved in the UPR signaling cascade [7]. GRP78 maintains the vital balance between folding peptides and the detaining of the ERS-generated UPR sensors. Recent studies showed that GRP78 is a multifunctional protein with activities far beyond its well-known role in the UPR, implicated in promoting tumor proliferation, metastasis, drug resistance, and viral entry [8,9,10]. Under pathophysiological conditions, GRP78 can translocate from the ER to the cell surface acting as a coreceptor for various signaling molecules, as well as for viral entry [11]. GRP78 could be modified with polyubiquitylation for subsequent degradation through the ubiquitin proteasomal system, leading to the suppression of cell migration and invasion [12]. Since GRP78 is a master regulator in the ER, changes in its expression, activation, or inhibition have been associated with major diseases, such as cancer, cardiovascular disease, and neurodegenerative disease [8,13,14]. 

Silkworm *B. mori*, the domestic lepidopteran model insect, is an important economic insect for silk production, and the new biotechnological methodology and knowledge are increasing [15]. Having short generation time, a clear genetic background, rich genetic resources, and a considerable number of genes that are highly homologous to human genes, the silkworm has been widely used as an ideal model organism in various life science studies [16]. In addition, due to long-term artificial domestication for its fine cocoon filaments, its resistance to adverse potential environmental pollution and various microbial pathogens in the natural habitat has become increasingly weak, making it susceptible to conditions of ER stress [17,18]. The previous study indicated that BmGRP78 played a possible central role in the silkworm secretion pathway [19]. However, the involvement of GRP78 in insects, particular in lepidopteran insects, compared with numerous works on mammalian GRP78s [10,20], remains largely unexplored. Therefore, we explored the molecular characterization and the response of GRP78 under biotic and abiotic stresses. 

In this study, we identified the molecular characterization of GRP78 from the silkworm *B. mori* and investigated its expression patterns. In addition, the chaperone characteristics, induced-expression, and subcellular localization were also analyzed. These results showed that BmGRP78 is likely to play an important role in in the process of environmental and physiological stimulation in silkworm. Thus, these results will lay a foundation for further understanding of the functions of GRP78 in lepidopteran insects. 

## 2. Results

### 2.1. Cloning and Characterization Analysis of BmGRP78 from Bombyx mori 

We isolated a 1977 bp cDNA fragment of *BmGRP78* from *B. mori*, encoding a protein of 658 amino acids. The deduced amino acid sequence comprised a predicted molecular weight of 73 kDa and a theoretical pI of 5.12. The amino acid sequence analysis showed that BmGRP78 contained NBD (from 31 to 409 aa), SBD (from 412 to 568 aa), and C-terminus (from 562–644 aa) (Figure 1A). Additionally, sequence analysis revealed that BmGRP78 contained three conserved signature patterns for the HSP70 family protein and carried the ER retention signal KDEL at its C-terminus (Figure 1B), which allowed for speculation that BmGRP78 was an Hsp70 family chaperone, localized in the ER lumen. The tertiary structure of BmGRP78 indicated that it could fold to form mature NBD and SBD domains (Figure 1C), and the ATP molecule can bind with the NBD domain of BmGRP78 (Figure 1D). These results indicated that BmGRP78 had the similar structure with the other species GRP78s.

Multiple sequence alignment of BmGRP78 further confirmed the presence of the NBD and SBD domains, which were highly conserved domains of GRP78 across species. The deduced amino acid sequence of BmGRP78 shared a high similarity with its counterparts in other species. BmGRP78 shared an 88.85%, 83.79%, 80.18%, 80.03%, and 79.13% identity with that of *Drosophila melanogaster*, *Penaeus chinensis, Rattus norvegicus*, *Homo sapiens*, and *Xenopus laevis*, respectively (Figure 1B). The phylogenetic analysis revealed that BmGRP78 clustered with the GRP78 of other invertebrates and vertebrates, and that it possesses a closer relationship with the GRP78 of *D. melanogaster* (Figure 2A,B), indicating that BmGRP78 is a homolog of the reported species GRP78s.

### 2.2. Expression and Production of Recombinant BmGRP78 (rBmGRP78) and Preparation of Antiserum 

The examination of the chaperone activity and detection of the endogenous expression of BmGRP78 required the preparation of the recombinant protein of BmGRP78. The complete ORF of *BmGRP78* was amplified by PCR and inserted into the expression vector. Following the transformation of *E. coli* BL21 (DE3) with the pET-32a(+)-*BmGRP78* construct, the *E. coli* culture was treated with IPTG to induce rBmGRP78 expression. The predicted molecular weight of BmGRP78 is approximately 73 kDa, and the fused label deriving from the pET-32a(+) is about 20 kDa. Thus, SDS-PAGE analysis revealed the recombinant protein with a molecular weight of approximately 100 kDa (Figure 3A and Appendix A), which was consistent with the predicted size of rBmGRP78. After purification with Ni-NTA His Bind Resin, a relative single band of the rBmGRP78 was detected using SDS-PAGE (Figure 3B). The purified rBmGRP78 protein was used for the preparation of rabbit antiserum. Anti-BmGRP78 polyclonal antibody was generated in an immunized New Zealand rabbit. The anti-rBmGRP78 polyclonal antibody titer, as determined by ELISA, was 1:256,000, and Western blotting analysis indicated that the antibody reacted specifically with rBmGRP78, BmN cells, and epidermis tissue, respectively (Figure 3C). These results indicated that the rBmGRP78 and the anti-rBmGRP78 antibody were successfully prepared for the following experiments.

### 2.3. Analysis Expression Patterns of BmGRP78 in Developmental Stages and Larval Tissues 

The expression profiles of the *BmGRP78* gene were analyzed in different developmental stages and various tissues of the third day of the fifth instar larvae (L5D3) of *B. mori* by qRT-PCR and Western blotting. The results showed that the *BmGRP78* gene was ubiquitously expressed in different developmental stages (Figure 4A and Appendix A) and examined tissues (Figure 4B and Appendix A) of *B. mori*, with different expression patterns. *BmGRP78* transcription levels were more significant in the fourth instar larvae on the third day (L4D3) than at other developmental stages, and the lowest transcription level was found at the egg stage (Figure 4A). On the other hand, we observed that *BmGRP78* was the most abundant in the gonad, followed by the middle silk glands (MSG), and the midgut, whereas lower transcription levels presented in the epidermis, fat body, trachea, and posterior silk glands (PSG) (Figure 4B).

### 2.4. The ATPase Activity of rBmGRP78 

GRP78 protein act in ATP-dependent cycles of binding and releasing of client substrates. The ATPase activity of rBmGRP78 under temperatures of 20 °C to 70 °C was investigated in this section. The peak of the ATPase activity was at approximately 50 °C (Figure 5A), indicating that the optimal temperature for the ATPase activity of rBmGRP78 was about 50 °C. The ATPase activity of rBmGRP78 gradually increased with the increasing temperature from 20 to 50 °C, and it was more than 1.5 U/mg prot at 50 °C. However, when the temperature was higher than 50 °C, the ATPase activity gradually decreased with the increasing of the temperature. The ATPase activity was not significant at 70 °C. These results indicated that the ATPase activity of rBmGRP78 was very stable at high temperatures, exhibiting a high temperature dependence. 

### 2.5. rBmGRP78 Inhibits Thermal Aggregation of Citrate Synthase (CS) and Malate Dehydrogenase (MDH)

To assess the effects of BmGRP78 on inhibiting the aggregation of substrate, we determined the structure of the aggregating thermolabile model substrates, CS or MDH, subjected to heat-induced aggregation at 48 °C or 45 °C in the presence or absence of rBmGRP78, respectively. The conformational changes in CS or MDH were analyzed by SDS-PAGE, respectively. The incubation of CS at 48 °C for 20 min resulted in heat-induced aggregation, while inclusion of rBmGRP78 reduced the extent of the heat-induced aggregation of CS (Figure 5B). A similar result was shown when MDH used as the targeted protein was employed in this assay (Figure 5C). The incubation of MDH at 45 °C resulted in heat-induced aggregation after 20 min; however, the addition of rBmGRP78 at a 1:2 molar ratio resulted in a reduction in aggregation. These results indicated that rBmGRP78 could inhibit the thermal aggregation of the thermolabile model substrates CS and MDH. 

### 2.6. BmGRP78 Expression in Response to Abiotic and Biotic Stresses in BmN Cells

To explore the involvement of BmGRP78 in responding to abiotic stresses, BmN cells were exposed to both heat and heavy metal stressors at various time points. The expression patterns of *BmGRP78* were determined using qRT-PCR and Western blotting. *BmGRP78* mRNA expression presented distinct patterns among heat, Pb, or Hg treatments (Appendix A), while BmGRP78 expression was consistently remarkably increased at the translational level (Figure 6A–C,E–G). Moreover, we observed that the BmGRP78 translational level was significantly induced at approximately 4 h, 6 h, and 6 h after heat, Pb or Hg treatments, respectively. To evaluate whether BmGRP78 was affected by biotic stresses, BmN cells were infected with BmNPV at various time points. After infection, BmGRP78 expression was not induced in BmN cells, either at the transcriptional or translational levels (Appendix A and Figure 6D,H). Thus, different BmGRP78 expression patterns existed in response to different stresses in BmN cells. 

### 2.7. Subcellular Localization of BmGRP78 under Different Stress-Induced Conditions in BmN Cells 

The subcellular localization of proteins can provide useful insights about their functions and help in understanding the intricate pathways that regulate biological processes at the subcellular level [21]. First, the subcellular location of BmGRP78 predicted by bioinformatics analyses showed 88.9% located in the endoplasmic reticulum and 11.1% located in extracellularly, including in the cell wall. We also used immunofluorescence staining and created an EGFP-BmGRP78 fusion protein to observe its subcellular location under abiotic and biotic stresses using laser confocal microscopy (Figure 7 and Appendix A). Consistent with the prediction, the BmGRP78 protein was mainly observed in the cytoplasm of untreated BmN cells. However, more BmGRP78 protein entered into the nucleus of the BmN cells at different levels when exposed to heat, heavy metal (Pb or Hg), or BmNPV treatments compared with those in untreated cells. These results suggested that BmGRP78 might play a key role in BmN cells under different stress-induced conditions. 

## 3. Discussion

As a highly conserved ER resident protein, GRP78 is involved in the folding and assembly of nascent proteins, serving as a master regulator of ER stress responses, facilitating cancer cell growth and viral replication [8,9,22]. GRP78 has been extensively studied in mammals [8], *Drosophila* [23], and fish [24]. However, the information regarding its molecular function in insect species is scarce. The present study identified the molecular characterization of the *GRP78* gene from *B. mori*, along with its expression patterns. The deduced amino acid sequence of BmGRP78 was found to be highly identical to that of other GRP78s, sharing conserved NBD and SBD domains. The signal peptide sequence identified in BmGRP78 indicated that BmGRP78 may not only target the ER, but may also serve as a secreted protein. Some studies have reported that GRP78 can escape to the cell surface upon ER stress, interacting with several ligands, and regulating cell signaling, apoptosis, and immune response [8,11,25]. Phylogenetic analysis revealed that BmGRP78 was the most similar to other known GRP78s. Collectively, these results indicated that BmGRP78 was a structural and functional homolog of identified GRP78, and it might play multiple biological roles in the silkworm. 

Gene expression patterns in different developmental stages and tissue distributions are important parameters for exploring their possible biological roles. Therefore, we analyzed the spatiotemporal expression patterns of BmGRP78 in *B. mori*. The BmGRP78 exhibited a ubiquitous expression profile in different developmental stages and various tissues of *B. mori*, with the L4D3 stage and the gonad showing the highest transcription level. The evidence suggests that the GRP78 protein is an essential regulator of developmental processes in insects, and the mechanism of regulation may vary among different life stages and insect species [23,26]. GRP78 is reportedly associated with male reproduction, is expressed on the surface of human and rat sperm, and is associated with calcium-binding protein [27]. The inhibition of GRP78 expression may affect sperm function through the PI3K/PDK1/AKT pathway and tyrosine phosphorylation in spermatozoa [28]. Thus, a wide spatiotemporal expression of the *BmGRP78* gene suggests that this protein may play a critical biological role in the process of development, as well as be involved in the spermatocyte growth regulation in *B. mori.* However, further studies are needed to verify its functions.

The chaperone activity of GRP78 depends on the binding and hydrolysis of ATP, and it uses ATP binding and hydrolysis at NBD to control the binding and release of client polypeptides at SBD [29]. The present study showed that BmGRP78 had highly conserved NBD and SBD domains, displaying properties including ATPase activity and the ability to inhibit the heat-induced aggregation of client proteins. These results show that GRP78 protein acts in the ATP-dependent cycles of the binding and releasing of client substrates, which are similar to the results of previous studies examining the properties of HSPs in *Drosophila* and mammalian model systems [30]. The chaperones are central to proteostasis, sharing highly conserved NBD and SBD domains that allosterically communicate in an ATP-dependent manner to recognize and bind client proteins [31].

ER-stresses trigger the UPR through multiple signaling pathways, including the upregulation of a family of chaperones called glucose-regulated proteins (GRPs) [32]. Growing evidences suggest that GRP78 generally serves as a chaperone to protect living organism against different extrinsic or intrinsic stimuli [33]. To understand this regulation under different extrinsic or intrinsic stimuli in BmN cells, we detected transcription and translation levels and the subcellular localization of BmGRP78 in response to various stresses. These results revealed that heat and heavy mental exposure induced its expression at the translation level, and BmGRP78 entered into the nucleus of BmN cells. Upon UPR, GRP78 changes its binding preference toward unfolded proteins, releasing the three sensors, including PERK, ATF6, and IRE1, to initiate UPR signaling [5,34]. After being released from GRP78, ATF6, is transported to the Golgi and proteolytically cleaves and releases the cytosolic N-terminus of ATF6 (N-ATF6), serving as a transcriptional factor to induce chaperone genes, including *GRP78* [35]. Under various stress conditions, GRP78 can also undergo functional change via PTMs, including sulfenylation, glutathionylation, ADP-ribosylation, phosphorylation, AMPylation, and citrullination, to regulate GRP78 activity, turnover, and availability [30,36]. Thus, we speculated that BmGRP78 expression at the transcriptional and translational level was not consistent because of post-translational modifications (PTMs). 

Apart from its role in protein folding, novel functions for GRP78 have also been described during infection [10]. It can also protect the immune system and the immune cells involved in the protection of cytoplasmic components, including all kind of biological factors, such as bacterial and viral infection [37]. Therefore, we also analyzed the expression of BmGRP78 in response to BmNPV infection. Our results showed that the BmNPV infection did not induce BmGRP78 expression in BmN cells, while more BmGRP78 protein entered into the nucleus of BmN cells compared with the untreated cells. Previous studies reported that human cytomegalovirus (HCMV) activated *GRP78* gene expression through direct promoter binding and the modulation of the local chromatin structure, which indicated an active viral mechanism of cellular chaperone induction for viral growth [35]. Overall, these results suggest that BmGRP78 might play a key role under different stress-induced conditions. However, the detailed mechanisms under different stimuli remain unknown, requiring further study at individual level.

In summary, we identified the molecular characterization and expression patterns of the GRP78 gene from *B. mori*. Its high conservation across species and ubiquitous expressed in different developmental stages and examined tissues of *B. mori* suggest that it may play multiple biological roles in the silkworm. As a molecular chaperone, the ATPase activity and the ability to inhibit the heat-induced aggregation of client proteins indicate that its activity is regulated by an allosteric ATPase cycle. Heat, Pb, Hg, and BmNPV exposure significantly induced its expression at the translation level or resulted in its translocation into the nucleus. However, further investigation is required to shed light on its biological mechanism at the cell and even individual level. Furthermore, to better understand BmGRP78’s contribution to cell survival in response to ER stress, the interplay between BmGRP78’s roles in UPR, autophagy, and apoptosis requires further investigation. 

## 4. Materials and Methods 

### 4.1. Biological Materials

The BmN was cultured at 27 °C in Sf-900™ medium containing 10% fetal bovine serum (Gibico, New Zealand) [38]. A multivoltine and non-diapausing silkworm strain (*Nistari*) was used in this study. The feeding temperature of silkworm larvae was 25 °C, and they were fed with fresh mulberry leaves under standard conditions, as described in previous studies [39]. *B. mori* nucleopolyhedrovirus (BmNPV) was stored in our laboratory [40]. 

### 4.2. Cloning and Sequence Analysis of BmGRP78 

The total RNA from the midgut tissue was extracted using the TRIzol reagent (Invitrogen, Carlsbad, CA, USA), and first-strand cDNA was synthesized using the PrimrScript^®^RT reagent kit (Takara, Beijing, China). The gene-specific primers (Table 1) were designed using the Primer Premier 5.0 software package based on the predicted nucleotide sequence of the *BmGRP78* gene (NM_001043372.1) from NCBI for the amplification of the *BmGRP78* cDNA fragment. The following amplification program was used for PCR: 4 min at 94 °C, followed by 35 cycles at 94 °C for 30 s, 55 °C for 60 s, and 72 °C for 2 min, followed by a final elongation step at 72 °C for 10 min. The PCR products were subjected to agarose gel (1%) electrophoresis, and the fragment of interest was purified and then sequenced (Sangon Biotech, Shanghai, China) after cloning into the pMD 19-T vector.

### 4.3. Bio-Informatic Analysis

The deduced amino acid sequence of BmGRP78 was obtained by using the online ExPASy program (http://www.expasy.ch/ accessed on 10 February 2021), and its domains were predicted using SMART (http://www.smart.embl-heidelberg.de/ accessed on 10 February 2021) and Interpro (http://www.ebi.ac.uk/interpro/ accessed on 10 February 2021). Blast analysis was performed on the NCBI website (http://blast.ncbi.nlm.nih.gov/Blast accessed on 10 February 2021) to determine BmGRP78 sequence similarities with other GRP78 proteins. Multiple sequence alignment was performed using the Clustal W program. The evolutionary tree was constructed by MEGA 7 using the neighbor-joining (NJ) method, with a bootstrap test of 1000 replications [41,42]. The three-dimensional structure of BmGRP78 was predicted using the online SWISS-MODEL [43,44] (https://swissmodel.expasy.org/interactive accessed on 10 March 2022). PyMOL (Schrödinger) was used in building the model for the domains of BmGRP78. 

### 4.4. Prokaryotic Expression, Purification, and Antiserum Preparation of Recombinant BmGRP78 (rBmGRP78)

The prokaryotic expression vector pET-32a(+)-*BmGRP78* was constructed and used to express His-fused BmGRP78 protein. Briefly, a pair of specific primers (Table 1) was designed to amplify the entire ORF of the *BmGRP78* gene containing the 1977 bp DNA fragment. The fragment of the *BmGRP78* gene was digested with restriction enzymes (*Hind*III and *Xho*I), sub-cloned into the pET-32a(+) expression vector, and then transformed into *E. coli* BL21 (DE3) (TransGen, China) for protein expression. The expression of rBmGRP78 protein was induced by IPTG (Solarbio, Beijing, China) and subsequently purified by Ni-NTA His Bind Resin (Novagen, USA), according to the manufacturer’s instructions. The purity of rBmGRP78 protein was determined by 12.5% sodium dodecyl sulfate-polyacrylamide gel electrophoresis (SDS-PAGE) and Coomassie Brilliant Blue R-250 staining.

Purified rBmGRP78 protein was used to prepare rabbit polyclonal antibody against BmGRP78 (HuaBio, Hangzhou, China), as described in previous studies [45]. The female New Zealand white rabbit was immunized with a subcutaneous injection of rBmGRP78 emulsified with Freund’s complete adjuvant (Sigma). The resultant antibody detected a single band of approximately 73 kDa on an immunoblot of protein extracted from BmN cells and the midgut. No bands were detected when pre-immune serum was employed in these experiments. The polyclonal antiserum was collected after the final immunization and stored at −80 °C for the following experiments.

### 4.5. RNA Extraction and Quantitative Real-Time PCR (qRT-PCR)

Quantitative real-time PCR (qRT-PCR) analysis was conducted to analyze the tissues and development expression profiles of *BmGRP78* in the silkworm strain, with minor modifications [46,47]. The gene-specific primers of *BmGRP78* and *Rp49* (endogenous control) were designed using Primer Premier 5.0 (Table 1). Total RNA was isolated from various tissues (head, midgut, fat body, epidermis, silk gland, Malpighian tubule, trachea, gonads, and hemolymph from the 3rd day of the 5th instar larvae) and different development stages of *B. mori* using TRIzol reagent (Invitrogen, Carlsbad, CA, USA), after grinding in liquid nitrogen, and then reverse transcribed into cDNA using the PrimrScript^®^RT reagent kit with gDNA Eraser (Takara, Beijing, China), following the manufacturer’s instructions. The relative expression of *BmGRP78* mRNA was quantified with the ABI 7500 Real-Time PCR system (Applied Biosystems, Foster City, CA, USA) using a SYBR^®^Premix Ex TaqTM Ⅱ kit (Takara, China). The endogenous *B. mori Rp49* gene (*BmRp49*, accession number: NM_001098282) was used as a normalizer. A relative quantitative method (threshold cycle [ΔΔCt]) was used to present the relative expression levels of the detected *BmGRP78* gene. All samples were tested with three independent replicates.

### 4.6. SDS-PAGE and Western Blotting 

The protein samples were prepared as follows. First, samples were dissolved in 100 µL PBS buffer, ground on ice for 5 min, incubated at room temperature for 30 min, and centrifuged at 15,000 rpm for 2 min at 4 °C twice, and the supernatant was extracted. The concentration was estimated in each protein sample by using an Enhanced BCA Protein Assay Kit (Beyotime, Shinghai, China), and the protein content in the sample was calculated. Each protein sample was diluted to a concentration of 2.5 μg/μL, added to the protein electrophoresis loading buffer (62.5 mM Tris-HCl, 2.5% SDS, 10% glycerinum, 0.5% bromophenol blue, 5% β-mercaptoethanol), and boiled for 5 min to denature the proteins. Then, 10 μL of the diluted sample was subjected to SDS-PAGE (12.5% polyacrylamide gel). The gel was stained with Coomassie blue (0.1% Coomassie brilliant blue R-250, 10% acetic acid, 50% methanol). Meanwhile, 20 μL of the diluted sample was subjected to SDS-PAGE to perform Western blotting analysis to verify the presence of the BmGRP78 protein. The proteins were transferred onto a polyvinylidene difluoride (PVDF) membrane using a semi-dry transfer-blot at 1 mA/cm^2^ for 90 min. The PVDF membrane was blocked in TBST (0.136 M NaCl, 20 mM Tris–HCl pH 7.6, 0.1% Tween 20) containing 5% defatted milk powder for 1 h with gentle shaking at room temperature, washed with TBST three times, and then incubated with primary antibody (rabbit anti-BmGRP78 with 1:500) for 2 h at room temperature. After washing, the membrane was incubated with a 5000-fold horseradish peroxidase (HRP)-conjugated Affinipure Goat Anti-Rabbit IgG(H+L) antibody (SA00001-2, Proteintech, Wuhan, China) for 1 h at room temperature. After washing with TBST 3 times, the PVDF membrane was detected with an ECL luminescence reagent using a fluorescence and chemiluminescence imaging system (Tanon 5500, Shanghai, China). 

### 4.7. The ATPase Activity Assay

The ATPase activity assay of BmGRP78 was investigated using the ATPase Assay Kit purchased from Nanjing Jiancheng Bioengineering Institute (Nanjing, China) [48], which measures the abundance of free inorganic phosphate (Pi) released from the ATP hydrolysis reaction, forming a chromogenic complex and yielding an absorption peak at 636 nm. The reaction time was 20 min, with a wide temperature range (20–70 °C), according to the instructions.

### 4.8. Thermal Aggregation Assays 

To further test the chaperone function of BmGRP78, heat-induced aggregation assays were performed as described in previous studies [49,50], with several modifications. Briefly, citrate synthase (CS) or malate dehydrogenase (MDH) in 20 mmol/L Tris-HCl buffer (pH 8.0) was denatured for 20 min at 48 °C or 45 °C, respectively, in the presence or absence of rBmGRP78 with a 1:2 ration, followed by 15 min of centrifugation (14,000 r.p.m., 4 °C). The supernatant and precipitate were analyzed by SDS-PAGE, followed by Coomassie staining to verify the state of CS and MDH, respectively.

### 4.9. Abiotic and Biotic Treatment 

The BmN (BmN-4) cell was cultured as described above. Stress-induced BmGRP78 expression profiles were observed at different time points and treated with heat, heavy metal (Pb or Hg), and BmNPV, respectively. For heat treatment, the cells were exposed to high temperature (39 °C), then sampled at 0, 2, 4, 8, 12, and 24 h after heat-induction. For heavy metal treatment, the cells were exposed to Pb- (700 µg/L) or Hg- (500 µg/L) treated medium for 0, 3, 6, 9, 12, 24, 48, and 72 h, respectively. For BmNPV infection, cells were infected for 0, 6, 12, 24, 48, and 72 h. The concentrations of Pb or Hg were selected based on the results of empiric acute and chronic toxicity tests in our laboratory. The cells were collected in 3 biological replicates, frozen quickly in liquid nitrogen, and then stored at −80 °C for the experiments. 

### 4.10. Subcellular Localization by Confocal Microscopy 

We investigated the subcellular location of BmGRP78 under different stress-induced conditions using immunofluorescence staining [51,52] and green fluorescence protein (EGFP)-tagged cloning [53], as in previous studies, with some modifications, respectively. Briefly, BmN cells were cultured in laser confocal dishes (Solarbio, Beijing, China) at 70–80% confluency (2 × 10^6^ cells/well) and incubated with purified anti-BmGRP78 IgG for 2 h at room temperature in the dark. After rinsing three times in PBST, the cells were incubated with Cy3-labeled goat antirabbit IgG (SA00009-2, Proteintech, Wuhan, China). For EGFP-tagged clone, the BmGRP78 open reading frame was amplified and cloned into the pIEx-1-EGFP vector. The clone containing the appropriate insert segment was identified by sequencing. The positive construct (2 µg) of the *BmGRP78* gene was transfected into the BmN cells using Fugene 6 transfection reagent (Promega, Madison, WI, USA). After 24 h of incubation, the BmN cells were fixed with 4% paraformaldehyde for 15 min. Subsequently, the nuclei of the BmN cells were stained with DAPI for 10 min and observed under a confocal microscope (FV1200, Olympus, Tokyo, Japan). 

### 4.11. Statistical Analysis 

The statistical analysis was performed using SPSS software (version 19, IBM Company, Armonk, NY, USA). All data were expressed as mean ± standard error (S.E.) of three independent measurements. Statistical comparisons were performed by one-way ANOVA analyses for testing the significance. Differences with *p* < 0.05 were considered statistically significant compared with the control group.

## Figures and Tables

**Figure 1 ijms-24-03964-f001:**
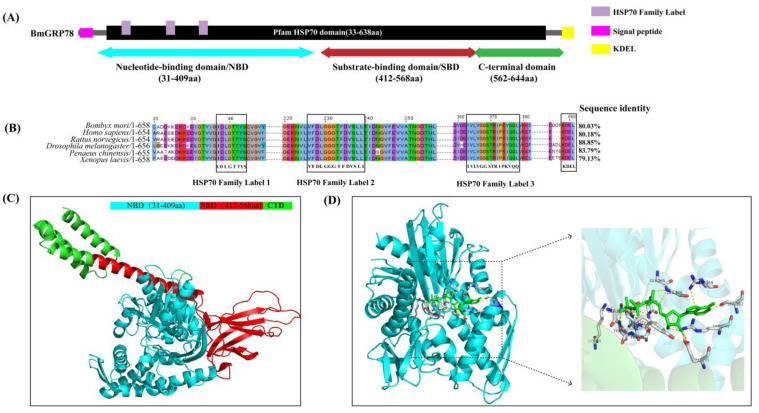
The domains, comparison, and predicted three-dimensional structure of the BmGRP78. (**A**) The upper panel represents the primary structure of BmGRP78, including the nucleotide binding domain (NBD), substrate binding domain (SBD), and C-terminal domain. In addition to the NBD and SBD, BmGRP78 also has an amino-terminus signal sequence, which is a 20-amino acid long signal peptide for ER targeting and translocation, and a carboxy-terminus ER retrieval sequence, lys-asp-glu-leu (KDEL), which prevents BmGRP78 from being secreted from the ER. (**B**) The middle panel shows a comparison of the BmGRP78 sequence of the species. (**C**) The predicted three-dimensional structure of the three domains in BmGRP78. The protein is represented by cartoons colored cyan (NBD), red (SBD), and green (C-terminal). The numbers represent the start and end of amino acid residues of the BmGRP78 domains. The models were generated by SWISS-MODEL, using Human GRP78 as the template (PDB ID 6asy1). (**D**) The NBD domain with the ATP molecule (PDB ID 5F1X) is represented by the green cartoon. The green stick represents the ATP that binds to the NBD of BmGRP78. The binding site with ATP is enlarged to show the amino acids involved in the interaction. The PyMol 2.4 software is used to represent the entire figure. (For interpretation of the references to color in this figure legend, the reader is referred to the web version of this article.)

**Figure 2 ijms-24-03964-f002:**
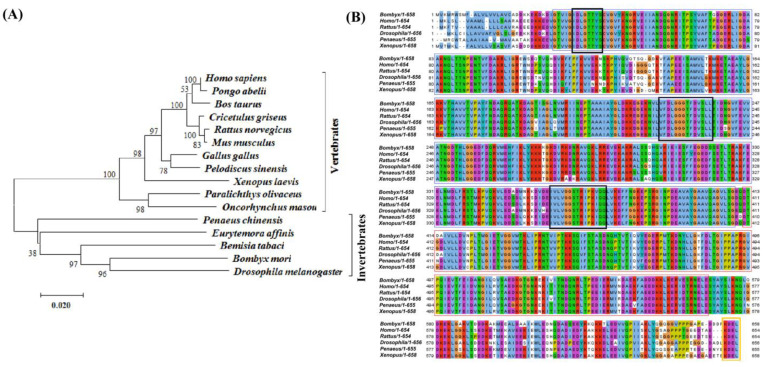
Phylogenetic analysis and sequence alignment of BmGRP78 protein with its homologs from invertebrate and vertebrate species. (**A**) The deduced amino acid sequences were aligned, and a phylogenetic tree was constructed by using MEGA (v7.0) and the neighbor-joining method. *Homo sapiens* (Accession No. NP_005338.1), *Rattus norvegicus* (Accession No. NP_037215.1), *Gallus gallus* (Accession No. NP_990822.1), *Mus musculus* (Accession No. NP_001156906.1), *Bos Taurus* (Accession No. NP_001068616.1), *Cricetulus griseus* (Accession No. NP_001233668.1), *Bemisia tabaci* (Accession No. ALP32533.1), *Bombyx mori* (Accession No. AGA84579.1), *Eurytemora affinis*, (Accession No. AEV42204.1), *Pelodiscus sinensis* (Accession No. ADW08701.1), *Paralichthys olivaceus* (Accession No. ABG56392.1), *Pongo abelii* (Accession No. NP_001126927.1), *Xenopus laevis* (Accession No. NP_001081462.1), *Oncorhynchus masou* (Accession No. BCK34772.1), *Drosophila melanogaster* (Accession No. NP_001285139.1), and *Pelodiscus sinensis* (Accession No. NP_001273821.1). (**B**) Alignment of the BmGRP78 protein with other homologous proteins. The deduced amino acid sequence of BmGRP78 was aligned with the GRP78 proteins from *Drosophila melanogaster* (NP_001285139.1), *Xenopus laevis* (NP_001081462.1), *Penaeus chinensis*, *Rattus norvegicus*, and *Homo sapiens*.

**Figure 3 ijms-24-03964-f003:**
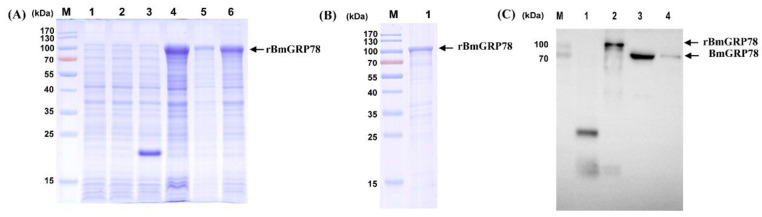
Identification of purified rBmGRP78 protein and specificity of rabbit polyclonal antibody against recombinant and endogenous BmGRP78 protein, respectively. (**A**) SDS-PAGE analysis for rBmGRP78 expression in *E. coli* BL21 (DE3) induced by IPTG. Lane M: Pre-stained Protein Marker; Lane 1: pET-32a(+) in BL21, no IPTG treatment; Lane 2: pET-32a(+)-*rBmGRP78* in BL21, no IPTG treatment; Lane 3: pET-32a(+) in BL21, IPTG treatment; Lane 4: pET-32a(+)-*rBmGRP78* in BL21, IPTG treatment; Lane 5: Supernatant of pET-32a(+)-*rBmGRP78* in BL21 after ultrasonication; Lane 6: Precipitation of pET-32a(+)-*rBmGRP78* in BL21 after ultrasonication. The arrow indicated the band of expected molecular weight for rBmGRP78 protein. (**B**) SDS-PAGE analysis of rBmGRP78 purification in *E. coli* BL21 (DE3) induced by IPTG. Lane M: Pre-stained Protein Marker; Lane 1, purified rBmGRP78 protein after ultrafiltration. (**C**) Identification of the specificity of anti-rBmGRP78 antibody using Western blotting. Lane M: Pre-stained Protein Marker; Lane 1, pET-32a(+) in *E. coli* BL21, no IPTG treatment; Lane 2, Supernatant of pET-32a(+)-*rBmGRP78* in BL21, IPTG treatment; Lane 3, Total proteins of BmN cells; Lane 4, Total proteins of epidermis tissue. The arrow represents recombinant and endogenous BmGRP78 protein.

**Figure 4 ijms-24-03964-f004:**
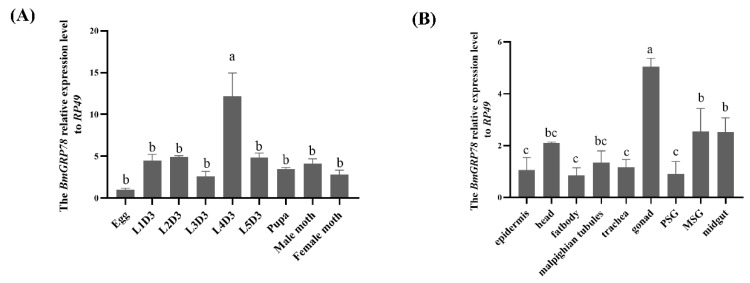
Expression patterns of *BmGRP78* in different developmental stages and tissues of *Bombyx mori.* The *BmGRP78* mRNA level in different developmental stages (**B**) and different tissues of L5D3 larvae (**A**) by qRT-PCR. Bars represent the mean ± S. E. (n = 3). Bars with different letters (a, b, and c) are significantly different (one-way ANOVA, followed by Tukey’s test, *p* < 0.05).

**Figure 5 ijms-24-03964-f005:**
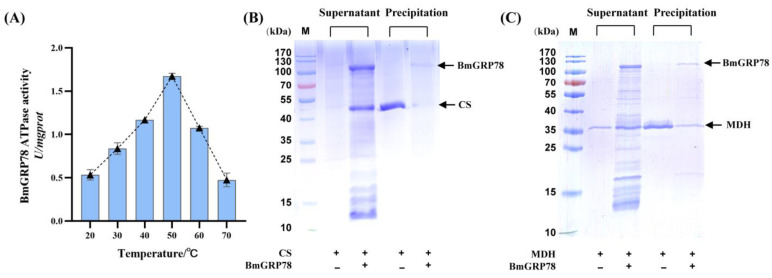
The ATPase activity and the prevention of the heat-induced aggregation of citrate synthase (CS) and malate dehydrogenase (MDH) by rBmGRP78 protein. (**A**) The ATPase activity of rBmGRP78 protein was measured from 20 °C to 70 °C. Bars represent mean ± S.E. (n = 3). The SDS-PAGE analyzed the aggregation of CS (**B**) and MDH (**C**), which was denatured for 20 min at 48 °C or 45 °C in the presence or absence of rBmGRP78, respectively.

**Figure 6 ijms-24-03964-f006:**
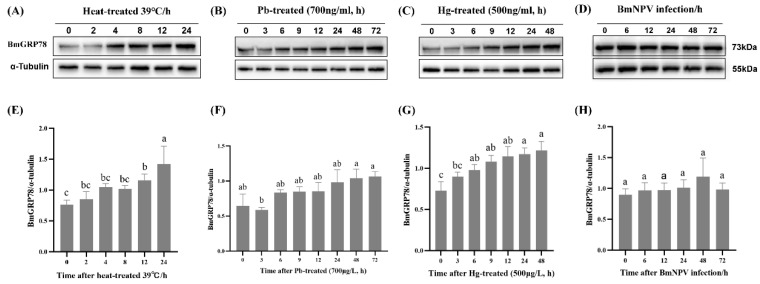
The translational level of BmGRP78 in response to different stress-induced conditions. (**A**,**E**) Heat-treated, (**B**,**F**) Pb-treated, (**C**,**G**) Hg-treated, (**D**,**H**) BmNPV infection. Bars represent mean ± S.E. (n = 3). Bars with different letters (a, b, and c) are significantly different (one-way ANOVA followed by Tukey’s test, *p* < 0.05).

**Figure 7 ijms-24-03964-f007:**
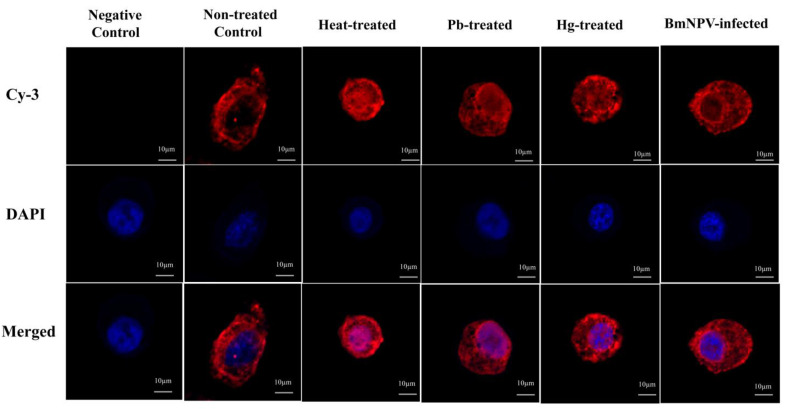
Subcellular localization of BmGRP78 in BmN cells under different stress-induced conditions. Subcellular localization of BmGRP78 with Cy3-labled secondary antibody and DAPI. The fluorescent signal was imaged by confocal microscopy. Immunostaining with anti-BmGRP78 antibody, DAPI blue fluorescence, merged images, and the negative control group are shown.

**Table 1 ijms-24-03964-t001:** Primer sequences used for gene cloning and verification in this study.

Gene	Primer Sequence (5′-3′)		Application	Length of Product (bp)
*BmGRP78*	Forward	ATGGTCAAGATGCG		
	Reverse	CAACTCGTCCTTGAAG		
	Forward	CCCAAGCTTGC**CACCACCACCACCACCAC**ATGGTCAAGATGCG	PCR	2012
	Reverse	CCGCTCGAGCAACTCGTCCTTGAAG		
	Forward	AAGGACATCGGCACAGTAATCG	qRT-PCR	141
	Reverse	ATCTTGAGTGAAGGCCACGTAT		
*RP49*	Forward	TGCTCCCAAATGGATTCCGTAAG	qRT-PCR	131
	Reverse	CACGATCAGCTTCCGCTTCTTC		

Note: *Hind*III and *Xho*I restriction sites are underlined. The blood sequence is His-tag.

## Data Availability

The data that support the results of this study are available within the article and the Supporting Information. Further data are available from the corresponding author upon reasonable request.

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
