# Peer review of "Biochemical Characterization and Functional Analysis of Glucose Regulated Protein 78 from the Silkworm *Bombyx mori"

_ijms, 2023, doi:10.3390/ijms24043964_

Round 1
Reviewer 1 Report
It is well known that the molecular chaperone glucose regulated protein 78 (GRP78) plays important roles in the endoplasmic reticulum (ER). However, GRP78 is a poor prognostic signal as an autoantigen in many malignancies, including prostate cancer, melanoma, colorectal cancer, and ovarian cancer, and it is also aberrantly expressed on the cell surface of many distinct types of tumor cells. In addition to acting as a receptor for the Dengue and Coxsackie A9 viruses, cell surface GRP78 may potentially contribute to Zika virus infection.
The manuscript describes the biochemical characterization and analysis of GRP78 from silkworm, Bm
Eight of the 21 references used in the introduction are older than 5 years, it is advisable to take help from the latest references in building the introduction section.
There are not many studies that have given the positive value of the GRP78, however, authors explored this protein from Bm expecting its positive outcome. How authors believe that this new source GRP78 is devoid of those issues as given in the literature.
Is the outcome sufficient to take the study forward on animal or clinical studies, is not clear. I mean do you have any data that neutralizes the problems and issues that are given in the literature for GRP78?
a,b,c used as abbreviations in Figure 4 (A) and (B) are not explained in the legend. Further, these details are even missing in figure 6 legend as well.
Reviewer 2 Report
In the present study, Xiao et al. characterized the GRP78 protein from the silkworm, B. mori. This work was performed on their previous finding that GRP78 was significantly up-regulated expression in the silkworm Nd mutation proteome database. GRP78 is an important chaperone for various environmental and physiological stimulation. The results and conclusions in this manuscript provide significant advances in this gene and may be useful for further studies on molecular mechanisms related to GRP78 in silkworms. Overall, the manuscript is well written and the experiments are correctly conducted, however, there are still a few points that need to be improved before it is acceptable for this journal.
1. The previous findings related to BmGRP78, including the papers published by authors, should be mentioned in the introduction section.
2. The manuscript investigated the expression and subcellular localization of BmGRP78 in response to abiotic and biotic stresses, which is very interesting and useful to make a better understanding of the function of this gene. However, the only functional experiments were conducted only in cultured cells. It would be better to add an investigation of the responses of BmGRP78 at the whole animal level.
Reviewer 3 Report
The topic is very interesting and opens up many possibilities for the advancement of molecular research on GRP78. Although the work is complex and features many experiments, the minor improvements are needed.
1) First of all, in the references section, volumes must be written in italics, as required by the journal.
2) On the line 41, The acronym HSP should be added.
3) It is important to give the brand and code specifications of the antibodies used. Add this information in lines 412 and 449.
